# Common Environmental Effects on Quantum Thermal Transistor

**DOI:** 10.3390/e24010032

**Published:** 2021-12-24

**Authors:** Yu-Qiang Liu, Deng-Hui Yu, Chang-Shui Yu

**Affiliations:** 1School of Physics, Dalian University of Technology, Dalian 116024, China; lyqmyt@163.com (Y.-Q.L.); ydhteam@163.com (D.-H.Y.); 2DUT-BSU Joint Institute, Dalian University of Technology, Dalian 116024, China

**Keywords:** open quantum system, heat currents, common environmental effects, quantum transistor

## Abstract

Quantum thermal transistor is a microscopic thermodynamical device that can modulate and amplify heat current through two terminals by the weak heat current at the third terminal. Here we study the common environmental effects on a quantum thermal transistor made up of three strong-coupling qubits. It is shown that the functions of the thermal transistor can be maintained and the amplification rate can be modestly enhanced by the skillfully designed common environments. In particular, the presence of a dark state in the case of the completely correlated transitions can provide an additional external channel to control the heat currents without any disturbance of the amplification rate. These results show that common environmental effects can offer new insights into improving the performance of quantum thermal devices.

## 1. Introduction

Quantum thermodynamics, which incorporates classical thermodynamics and quantum mechanics, has attracted wide attention [1,2,3]. It provides important theories to study thermodynamical quantities like heat, work, entropy, and temperature, or the thermodynamical behaviors in the microscopic world, while quantum thermal machines are significant subjects in quantum thermodynamics. The research on quantum thermal machines allows us not only to test the basic laws of thermodynamics in the quantum level, but also to exploit microscopic thermodynamic applications in terms of quantum intriguing features. Up to now, a great deal of efforts have been paid for the relevant topics [4,5,6,7,8,9,10,11,12,13,14,15,16,17,18,19,20,21,22,23,24,25,26,27,28,29,30,31,32,33,34,35,36], especially based on various working substances, such as two-level systems [12], multi-level spin systems [15,16,17,18], superconducting qubits [9,13,19], quantum dots [14], quantum harmonic oscillators [22], opto-mechanical systems [20,23], and so on.

Quantum self-contained thermal devices are one family of the most compelling thermal machines due to the small dimension of the quantum system and no external work or control resources. The self-contained thermal devices were originally proposed as refrigerators to cool the cold bath [12,13,14,25,26,27,28,29,30,31,32,33,34,35,36,37,38,39,40,41,42,43,44,45], or as thermal engines to extract work [46,47]. Later, they were widely extended to a variety of cases, including the different interaction mechanisms, or for achieving various functions, such as heat current amplification [9,24,48,49,50,51,52,53,54,55], thermal rectification [19,20,21,54,55,56,57,58,59,60], thermal batteries [61,62,63], and thermometers [64,65,66]. It is a quite fundamental question for quantum thermodynamics as to whether the performance of thermodynamic devices could be improved by some particular designs, such as measurement [23,67], or any quantum effects, such as quantum entanglement [38,39], quantum coherence [40,41,68,69,70,71], anharmonicity [72], non-Markovian effects [73], and so on, which have brought novel insights to the related research.

Recently, it has been shown that the common environmental effect can not only lead to the decoherence-free subspace [74,75], but also has significant contributions to quantum features [37,74,76,77,78,79]. Especially, it has been shown that the cooling power of a weak internal coupling refrigerator can be enhanced by common environments [37]. It brings some potential contributions to the performance of quantum thermal devices.

In this paper, we study the common environmental effects on the thermal transistor which were originally proposed in [49]. We let the three qubits composing the thermal transistor be commonly coupled to three reservoirs in the same manner of [37]. We show that the system can still work as a thermal transisor and the common environments influence the three heat currents of the thermal transistor to different degrees, so the amplification effect proportional to the ratio of thermal currents can be enhanced by appropriately designing the common coupling strengths. In particular, there exists a dark state in the system in the case of the completely correlated transitions. Since the dark state does not undergo the evolution and especially does not affect the amplification rate, it provides a significant channel to control the heat currents. We also revealed that the physical essence of the enhancement is the strengthened asymmetry induced by the common environmental effects. The remainder of the paper is organized as follows. In Section 2, we introduce the physical system of the thermal transistor and derive the master equation. In Section 3, we solve the dynamics and demonstrate the functions of the thermal transistor. In Section 4, we study the common environmental effects in detail and present the physical root. The conclusions and discussion are given in Section 5.

## 2. The Model of the Thermal Transistor

The working substance of the thermal transistor is three interacting qubits, labelled by *L*, *M*, *R*, respectively, which are in contact with three common reservoirs with the temperature given by $T_{L}$, $T_{M}$, $T_{R}$. The sketch of the whole system is shown in Figure 1. The Hamiltonian of the system reads (ℏ=1)
(1)$$H_{S}=\frac{1}{2}\sum_{\nu =L,M,R}\omega_{\nu }\sigma_{\nu }^{z}+g\sigma_{L}^{x}\sigma_{R}^{x}\sigma_{M}^{x},$$
where $\omega_{\nu }$ are the frequency of the qubit ν, $\sigma^{z/x}$ denote the Pauli matrix, and *g* is the coupling strength between the three qubits. In addition, the frequencies satisfy the relation $\omega_{R}=\omega_{L}+\omega_{M}$. It is obvious that we have kept the counter-rotating wave terms in the Hamiltonian due to the strong internal interaction. Note that the similar three-body interaction has been widely used in the relevant researches on the self-contained refrigerator [13,29,30,31,34,36,37,44]. In particular, some experimental proposals have been presented in [13,31,36], and the experimental realizations are reported in [33] and on the tripartite Greenberger-Horne-Zeilinger state in [80]. The Hamiltonian of the three reservoirs is given by
(2)$$H_{R}=\sum_{\nu ,l}\omega_{\nu l}b_{\nu l}^{†}b_{\nu l},$$
where $b_{\nu l}$, $b_{\nu l}^{†}$ denote the annihilation and creation operators of the reservoir modes with $\left[b_{\nu l},\;b_{\nu l}^{†}\right]=1$ and $\omega_{\nu l}$ denote the frequency of the reservoir modes. Suppose the three-qubit system interacts with the three reservoirs so weakly that we can safely employ the rotating wave approximation, then the interaction Hamiltonian between the system and the reservoirs can be written as
(3)$$H_{SR}=\sum_{\nu ,k}\left(f_{\nu j}S_{\nu }⊗b_{\nu l}^{†}+f_{\nu j}^{*}S_{\nu }^{†}⊗b_{\nu l}\right),$$
where $f_{\nu j}$ represent the coupling strength between the qubit ν and the *j* mode of its corresponding reservoir, in particular, the jump operators $S_{\nu }$ of the νth system are taken as the same form as [37]:(4)$$\begin{array}{c} S_{L}=\sigma_{L}^{-}+\lambda_{1}\sigma_{R}^{-}\sigma_{M}^{+},\\ S_{M}=\sigma_{M}^{-}+\lambda_{2}\sigma_{L}^{+}\sigma_{R}^{-},\\ S_{R}=\sigma_{R}^{-}+\lambda_{3}\sigma_{L}^{-}\sigma_{M}^{-}, \end{array}$$
with the parameters $\lambda_{1},\lambda_{2},\lambda_{3}\in \left[0,1\right]$. As stated [37,81], one or two-spin simultaneous transition of the system can be induced by a single excitation in the environment. Obviously, these jump operators include the correlated single- and double-qubit transitions, where $\lambda_{1}=\lambda_{2}=\lambda_{3}=0$ means that each qubit is in contact with its independent reservoir, while $\lambda_{1}=\lambda_{2}=\lambda_{3}=1$ implies that the completely correlated transitions induced by the common reservoirs. Thus, the Hamiltonian of the whole composite system including the system and reservoirs is
(5)$$H=H_{S}+H_{R}+H_{SR}.$$
The Hamiltonian Equation (Equation 1) can be written in the eigen-decomposition form as
(6)$$H_{S}=\sum_{i}\epsilon_{i}|\epsilon_{i}\rangle \langle \epsilon_{i}|,$$
where $\epsilon_{i}$ are the eigenvalues and $|\epsilon_{i}\rangle$ are their corresponding eigenstates which are explicitly given in Appendix A. In the $H_{S}$ representation, the jump operators are given by
(7)$$S_{\nu ,k}=\sum_{\epsilon_{j}-\epsilon_{i}=\omega_{\nu ,k}}|\epsilon_{i}\rangle \langle \epsilon_{i}\left|S_{\nu }\right|\epsilon_{j}\rangle \langle \epsilon_{j}|,$$
where $S_{\nu ,k}$ explicitly given in Appendix B, is the eigen-operator subject to the commutation relation
(8)$$\left[H_{S},\;S_{\nu ,k}\right]=-\omega_{\nu ,k}H_{S},$$
with the eigen-frequency $\omega_{\nu ,k}$. It will be found that every $S_{\nu }$ induces four non-vanishing $S_{\nu ,k}$, k=1,2,3,4, which means that the system interacts with every heat reservoir via 4 channels. With the eigen-operators, Equation (Equation 3) can be rewritten as
(9)$$H_{SR}^{\prime }=\sum_{\nu ,k,l}\left(f_{\nu j}S_{\nu ,k}\left(\omega_{\nu ,k}\right)⊗b_{\nu l}^{†}+h.c.\right),$$
and the total Hamiltonian Equation (Equation 5) is rewritten as
(10)$$H=H_{S}+H_{R}+H_{SR}^{\prime }.$$

## 3. Dynamics of the System and the Thermal Transistor

In order to get the steady-state behavior of the transistor system, we have to begin with the dynamics of the system which is governed by the master equation. As mentioned previously, the three qubits of our thermal transistor are strongly coupled to each other. Therefore, we have to consider the three qubits as a whole and derive a global master equation under reasonable appoximations. It is fortunate that [82,83] have provided quite a standard process to arrive at the Lindblad master equation with the Born-Markov-secular approximation. Here we’d like to emphasize that we restrict our thermal transistor to the valid Born-Markov-secular approximation and strictly follow the standard process to derive the master equation. We have obtained the same form of master equation as the standard Lindblad master equation given in [82,83]. Thus, one can directly subsitute the eigen-frequencies and the eigen-operators given in Equation (Equation 8) and Appendix B into the standard Lindblad master equation [82,83] to obtain the dynamical equation of our current system, which is given as
(11)$$\begin{array}{cc} \dot{\rho }= & L_{L}\left[\rho \right]+L_{M}\left[\rho \right]+L_{R}\left[\rho \right],\\ L_{\nu }\left[\rho \right]=\sum_{\nu } & \gamma_{\nu }\left(w_{\nu ,k}\right)\left(\bar{n}+1\right)D\left[S_{\nu ,k}\right]+\gamma_{\nu }\left(w_{\nu ,k}\right)\bar{n}D\left[S_{\nu ,k}^{†}\right], \end{array}$$
where $\bar{n}\left(w_{\nu ,k}\right)=\frac{1}{e^{\frac{\omega_{\nu ,k}}{T_{\nu }}}-1}$ denotes the average number of photons of the mode with frequency $\omega_{\nu ,k}$ ($k_{B}=1$) and the Lindblad super-operator is defined as $D\left[x\right]=x\rho x^{†}-\frac{1}{2}\left\{x^{†}x,\rho \right\}$. As usual, we assume the reservoirs in the thermal equilibrium state $\rho_{\nu }=\exp\left(-H_{\nu }/T_{\nu }\right)/\mathrm{Tr}\left[\exp\left(-H_{\nu }/T_{\nu }\right)\right]$ and the spectral density of reservoirs $\gamma_{\nu }\left(w\right)=2\pi \sum_{k}$$|f_{\nu k}{|}^{2}\delta \left(w-\omega_{k}\right)$ with $\gamma_{\nu }\left(w_{\nu ,k}\right)=\gamma_{\nu }$ as a constant for simplicity. In addition, during the derivation we have employed the secular approximation, which requires the Bohr frequency differences to be much larger than the inverse of the reservoir correlation time $\tau_{R}$, that is, 2g≫γ for the current system with $\omega_{\nu }>g$. In the following calculations, we will always keep these conditions satisfied.

The master equation Equation (Equation 11) involves the evolution of both the diagonal and off-diagonal entries of the density matrix ρ in *H* representation. A detailed calculation can show that the diagonal entries evolve independently of the evolution of the off-diagonal entries and especially the off-diagonal entries will vanish in the steady-state density matrix. Therefore, we will only consider the evolution of the diagonal entries. Thus one can arrive at the differential equation for the diagonal entries $\rho_{kk}$, termed populations [84], as
(12)$${\dot{\rho }}_{kk}=\sum_{\nu =L,M,R}\sum_{l}T_{kl}^{\nu }\left(\rho \right),$$
where
(13)$$T_{ij}^{\nu }\left(\rho \right)=\gamma_{\nu }\left[\left(\bar{n}\left(\omega_{ij}\right)+1\right)\rho_{jj}-\bar{n}\left(\omega_{ij}\right)\rho_{ii}\right]{\left|\langle \epsilon_{i}|S_{\nu }|\epsilon_{j}\rangle \right|}^{2}$$
with $\omega_{ij}=\epsilon_{j}-\epsilon_{i}$ describing the increment rate of the population $\rho_{kk}$. $T_{kl}^{\nu }\left(\rho \right)$ vanishes for i>j, because ${\left|\langle \epsilon_{i}|S_{\nu }|\epsilon_{j}\rangle \right|}^{2}$ is the same as the coefficients covered in the eigen-operators and determines the allowable transitions. Solving ${\dot{\rho }}_{kk}=0$, one will obtain the steady-state solution of the diagonal density matrix denoted by $\rho^{S}$. The heat current [82] can be given by
(14)$${\dot{Q}}_{\nu }=\mathrm{Tr}\left\{{\hat{H}}_{S}L_{\nu }\left[\rho^{S}\right]\right\}=\sum_{kl}T_{kl}^{\nu }\left(\rho^{S}\right)E_{lk},$$
where $E_{lk}=\epsilon_{k}-\epsilon_{l}$ denotes the transition energy. The expanded expression of the heat currents is given in Appendix C. The positive heat current indicates heat flowing from reservoir into the system and the negative current means the opposite flowing direction. In addition, it is easily found from Equation (Equation 14) that the three heat currents fulfill ${\dot{Q}}_{L}+{\dot{Q}}_{R}+{\dot{Q}}_{M}=0$, which indicates the conservation relation between work, heat, and internal energy.

## 4. The Common Environmental Effects

*Thermal transistor*. Since the common reservoirs are taken into account, first we briefly demonstrate that the functions of a thermal transistor can be realized. Based on Equation (Equation 14), it can be found in [85] that with certain parameters, we can make the heat current ${\dot{Q}}_{M}$ weak enough, so that it can be used as the control terminal of a transistor to modulate the heat currents ${\dot{Q}}_{L}$ and ${\dot{Q}}_{R}$, which is explicitly demonstrated in Figure 2. In particular, the heat currents ${\dot{Q}}_{L}$ and ${\dot{Q}}_{R}$ can be so small that they can be thought to be zero to some reasonable approximation. In this sense, it works as a thermal switch. In the modulation process, the weak heat current ${\dot{Q}}_{M}$ is changed slightly, but ${\dot{Q}}_{L}$ and ${\dot{Q}}_{R}$ change greatly, which shows the amplification effects that can be well-characterized by the amplification factor defined as [86]
(15)$$\alpha_{L,R}=\frac{\partial {\dot{Q}}_{L,R}}{\partial {\dot{Q}}_{M}}=\frac{\frac{\partial {\dot{Q}}_{L,R}}{\partial T_{X}}}{\frac{\partial {\dot{Q}}_{M}}{\partial T_{X}}},$$
where X=M,R,L represents the control terminal of the thermal transistor. The amplification effect appears if the amplification factor $|\alpha_{L,R}|>1$, which is only determined by the absolute value because the sign of $\alpha_{L,R}$ indicates whether the change trends of ${\dot{Q}}_{L,R}$ and ${\dot{Q}}_{M}$ are the same or not. From Figure 2, one can find that the amplification factor $|\alpha_{L,R}|$ is about 30, which ensures the apparent amplification effect.

*Enhancement of amplification factor*. Next we will show that one can utilize the correlated transitions to enhance the amplification effects of the thermal transistor.

In Figure 3, we have plotted the amplification factor versus $T_{M}$ for different interaction strengths with common environmental effects. One can observe that the amplification factor $|\alpha_{L,R}|$ decreases with the increasing temperature $T_{M}$, especially if there is no common system–reservoir coupling. The amplification effect is more sensitive to the temperature $T_{M}$ for $g=0.7\;\omega_{M}$ than for $g=0.3\;\omega_{M}$. The most important result is that the amplification effect is enhanced when increasing the collective transition strength $\lambda_{1}$. This is different from the phenomenon found in [37] that the coefficient of performance of the refrigerator is nearly invariant with increasing $\lambda_{i}$. In Figure 3c, the common environmental effect has led to the amplification factor increasing with the temperature $T_{M}$. All the figures in Figure 3 indicate that the enhancement effects of the common environment become strong with the increase of the temperature $T_{M}$. Thus, it seems that the common environment additionally tends to stabilizing the amplification rate of the thermal transistor. However, one will find that the amplification effects cannot always be enhanced by arbitrarily designed common environments. In Figure 3c, the amplification factor versus the temperature $T_{M}$ with a fully common environmental effect has shown that the amplification effect is reduced in some regions of low temperature. In particular, when we decrease the parameter $\lambda_{1}$, the suppression of the amplification rate will be more apparent than that in Figure 3c, which can be seen from the enlarged suppression region in Figure 3d. Therefore, in order to boost the amplification effect, one will have to design the common environments on purpose. Namely, we should not equally increase the common couplings $\lambda_{1}$, $\lambda_{2}$, $\lambda_{3}$. In addition, the quantum thermal transistor is a three-terminal thermodynamic device. Each terminal can be used as the control terminal. In Figure 4a,b, we have illustrated that the common environments play the enhancement effects if employing the terminal *L*, or *R* as the control terminal.

To demonstrate the best enhancement by the common environments, we optimize the amplification factor on the common couplings $\lambda_{\alpha }$, which is depicted in Figure 5. It is obvious that the common couplings $\lambda_{1}$ and $\lambda_{3}$ play positive and negative roles in the optimal enhancement, respectively. The maximal amplification factor can only be achieved by the particular $\lambda_{2}$. This is also consistent with those implied in Figure 3.

An intuitive understanding of the enhancement of the amplification effects can be given by analyzing the heat currents given in Equation (Equation 22) and the transitions. It can be seen that the heat currents are determined by the transition rates $T_{ij}^{\nu }\left(\rho \right)$ and the transition energies $E_{ij}$. The common couplings only slightly affect the populations of the systems, as shown in Figure 6 and modestly affect some of the eigen-operators. Even though the transition energies $E_{ij}=\epsilon_{i}-\epsilon_{j}$ are not influenced, they act weight-like on and much larger than the changes of $T_{ij}^{\nu }\left(\rho \right)$. The transition rates with large transition energies in heat current ${\dot{Q}}_{M}$ have no relation with the common coupling and other changed transition rates are only subject to the small transition energy, as a result, ${\dot{Q}}_{M}$ is slightly changed. In contrast, the large transition energies covered in the heat currents ${\dot{Q}}_{L}$ and ${\dot{Q}}_{R}$ greatly amplify the changes on the transition rates and induce the apparent effects on ${\dot{Q}}_{L}$ and ${\dot{Q}}_{R}$. Therefore, the amplification factor is significantly enhanced. Since ${\dot{Q}}_{L}$ and ${\dot{Q}}_{R}$ depend on $\lambda_{1}$ and $\lambda_{2}$, respectively, $\lambda_{1}$ and $\lambda_{2}$ naturally play the dominant roles. Therefore, roughly speaking, the heat currents are dominantly determined by the transition energies, which are essentially based on the asymmetry of the transition frequencies subject to the different heat baths.

*External controllable heat modulator*. We would like to emphasize that for $\lambda_{1}=\lambda_{2}=\lambda_{3}=1$ the eigenstate $|\epsilon_{4}\rangle$ is always a dark state, which is immune from the dissipation in the evolution. This can be easily found from Equations (Equation 19) and (Equation 20) because all the transitions in the eigen-operators related to $|\epsilon_{4}\rangle$ vanish with the vanishing $a_{\mu ,k}^{-}$. This indicates that $\rho_{44}^{S}=\rho_{44}\left(0\right)$ is determined by the initial state [43,87]. In this case, a simple algebra will show that there is no heat current between the three reservoirs for $\rho_{44}^{S}=\rho_{44}\left(0\right)=1$, which means the three reservoirs are isolated from each other. A detailed demonstration of the dependence on $\rho_{44}\left(0\right)$ is plotted in Figure 7a, which indicates that the heat currents monotonically decreases with $\rho_{44}\left(0\right)$. In this sense, one can control the magnitude of the heat currents by the initial state. The initial-state dependent heat current is also shown in the Ref. [37]. In addition, the amplification rate of the thermal transistor does not depend on the population $\rho_{44}\left(0\right)$ shown in Figure 7b. This can be well-understood as follows. Any initial state can be given in the $H_{S}$ representation, which is divided into two evolution subspaces. One corresponds to the dark state $|\epsilon_{4}\rangle$ which does not evolve and contributes nothing to the heat currents, and the other corresponds to the other subspace spanned by the remaining eigenstates of $H_{S}$, which will evolve to the steady state and has an active contribution to the heat currents. Thus, all the heat currents will be equally reduced by $\rho_{44}\left(0\right)$, so the amplification rate is not changed. This is also analytically demonstrated by Equation (Equation 31) in Appendix C.

To demonstrate that the heat currents can be well-modulated by $\rho_{44}$, we can suppose the following process. (i) First of all, the system is working as a transistor; (ii) at some particular moment $t_{0}$, one expects to adjust the heat currents. To achieve this task, one can first use a laser to drive the transition between $|\epsilon_{4}\rangle$ and $|\epsilon_{k}\rangle$ for some preset duration $\Delta t=t-t_{0}$ until $\rho_{44}$ reaches the expectation, then switch off the laser and let the system evolve to the steady system. Without loss of generality, let the interaction Hamiltonian of the driving be given by
(16)$$\begin{array}{cc} H^{D} & =\epsilon_{4}|\epsilon_{4}\rangle \langle \epsilon_{4}|+\epsilon_{8}|\epsilon_{8}\rangle \langle \epsilon_{8}|+\Omega (|\epsilon_{4}\rangle \langle \epsilon_{8}|e^{i\omega_{d}t}\\ & +|\epsilon_{8}\rangle \langle \epsilon_{4}|e^{-i\omega_{d}t}), \end{array}$$
where $\omega_{d}=\epsilon_{8}-\epsilon_{4}$ is the driving frequency and Ω denotes the driving strength. The evolution operator reads
(17)$$U\left(t\right)=e^{iH_{I}^{D}\Delta t},$$
with the interaction Hamiltonian $H_{I}^{D}=\Omega \left(|\epsilon_{4}\rangle \langle \epsilon_{8}|+|\epsilon_{8}\rangle \langle \epsilon_{4}|\right)$. The evolution of $\rho_{44}$ is plotted within one period in Figure 8a. Figure 8b is the steady-state heat currents under the given conditions, which can be understood as the initial working status of the thermal transistor. Figure 8c illustrates the steady-state heat currents after modulation. It is assumed that the modulation duration is $\Delta t=0.7\frac{\pi }{\Omega }$, which is determined by the expected heat currents. When reaching the steady state, the corresponding heat currents are shown in Figure 8c. Apparently, the heat currents have been greatly increased. Here we suppose that the driving is so short that the system evolves unitarily for simplicity.

## 5. Conclusions and Discussion

We have studied the common environmental effects on the quantum thermal transistor. It is found that the amplification rate of the thermal transistor can be raised by properly designing the common couplings with the environments. Due to the different enhancement effects with the temperature $T_{M}$, it seems that the common environments have the ability to stabilize the amplification rate. In addition, the enhancement is also present with different terminals as the control terminal. Analogous to [37], a dark state will occur in the case of the completely correlated transitions. This dark state can provide an additional channel to control the magnitude of the heat currents, but has no effect on the amplification rate. An intuitive physical understanding of the enhancement is also given in terms of the common environmental effect enhanced to different extents due to the asymmetry.

Finally, we would like to mention that the asymmetry as the physical root of the considered thermal transistor can be effectively controlled in many ways. The most intrinsic is the large detuning of the frequencies of the coupled qubits, whereas it could not be practical to infinitely enlarge the detuning. If the system-reservoir couplings are controllable, adding the proper bias to the decay rates γ could be a straightforward and usual method, the effects of which have been demonstrated in Figure A1a in Appendix D. The combined roles of the decay rate bias and the common environments are shown in Figure A1b, which indicates that the amplification rate can be raised much more. The common environments provide another alternative and novel approach to enhance the amplification rate; meanwhile, it could enrich the functions of a thermal device.

## Figures and Tables

**Figure 1 entropy-24-00032-f001:**
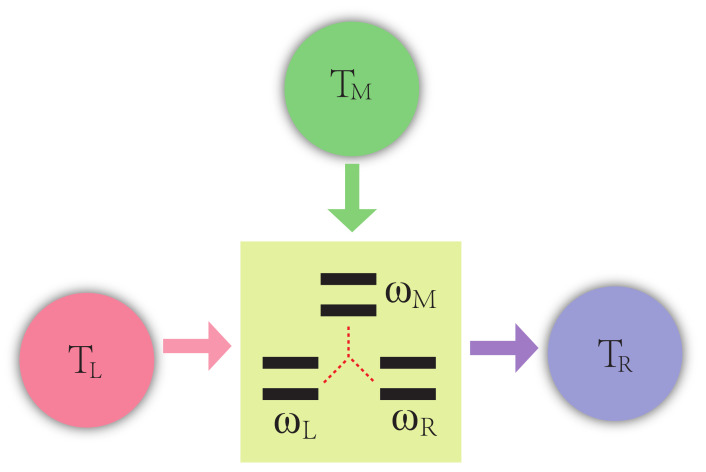
Sketch of the thermal transistor. The working substance of the system consists of three strong-coupling qubits with different frequencies $\omega_{L}$, $\omega_{M}$, $\omega_{R}=\omega_{L}+\omega_{M}$, respectively. The composite system interacts with three independent reservoirs at different temperatures $T_{L}$, $T_{M}$ and $T_{R}$ through different transition channels.

**Figure 2 entropy-24-00032-f002:**
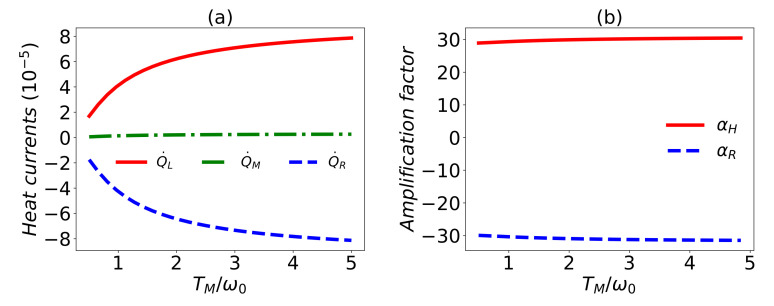
(**a**) Heat currents and (**b**) the amplification factor versus the temperature $T_{M}/\omega_{0}$. The weak heat current ${\dot{Q}}_{M}$ can lead to the great change of ${\dot{Q}}_{L}$ and ${\dot{Q}}_{R}$, which indicates the function of a thermal transistor. It is shown that the amplification factor $|\alpha_{L,R}|∼30$. In the figures, $\omega_{0}=1$, $\omega_{M}=\;\omega_{0}$, $\omega_{L}=30\;\omega_{0}$, $T_{L}=5\;\omega_{0}$, $T_{R}=0.5\;\omega_{0}$, $g=0.1\;\omega_{M}$, $\lambda_{1}=\lambda_{2}=\lambda_{3}=0.7$ and $\gamma_{L}=\gamma_{M}=\gamma_{R}=\gamma =0.002\;\omega_{M}$.

**Figure 3 entropy-24-00032-f003:**
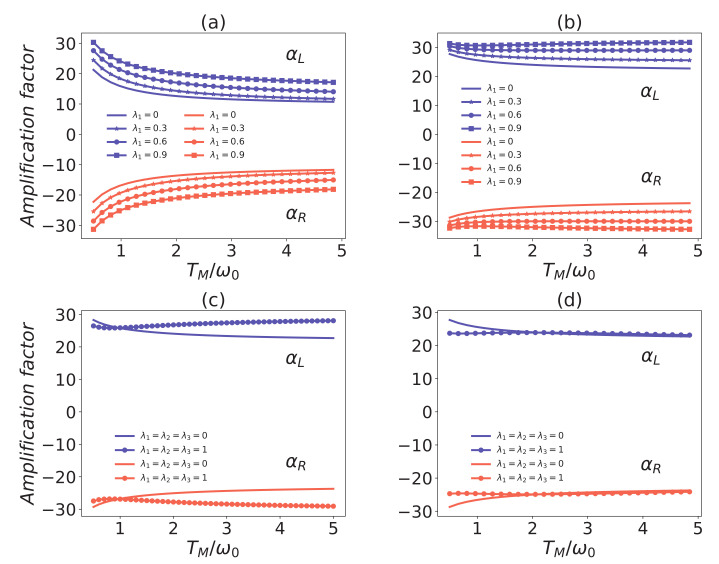
Comparison of amplification factors $\alpha_{L/R}$ versus the temperature $T_{M}/\omega_{0}$ with and without common environmental effect. All the solid curves correspond to the case without common environmental coupling. The coupling strength $g=0.7\;\omega_{M}$ in panel (**a**), and $g=0.3\;\omega_{M}$ in others panels. In panels (**a**,**b**), the parameters $\lambda_{2}=\lambda_{3}=0$ and $\lambda_{1}=0.3,\;0.6,\;0.9$ corresponds to the solid asterisk, the solid circle, and the solid square curves, respectively; In panel (**c**), the solid circle curve corresponds to completely common environmental couplings, that is, $\lambda_{1}=\lambda_{2}=\lambda_{3}=1$; In panel (**d**), $\lambda_{1}=0.2$, $\lambda_{2}=\lambda_{3}=0.8$ for the solid circle curve. The other parameters in all panels are the same as in Figure 2.

**Figure 4 entropy-24-00032-f004:**
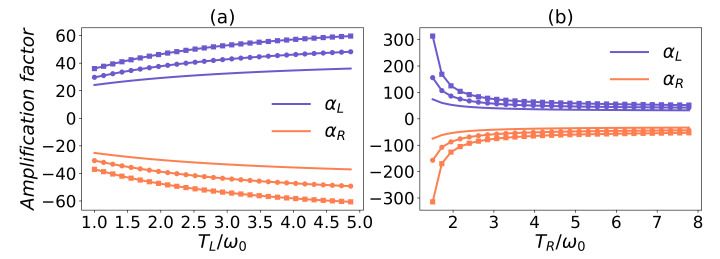
Amplification factor as a function of $T_{L}/\omega_{0}$ or $T_{R}/\omega_{0}$ with common environmental effect. In panel (**a**), $T_{M}=5\;\omega_{0}$, $T_{R}=1\;\omega_{0}$, and for panel (**b**), $T_{M}=7\;\omega_{0}$, $T_{L}=1.6\;\omega_{0}$. In all panels, $\lambda_{2}=\lambda_{3}=0.1$ and parameters $\lambda_{1}=0,\;0.45,\;0.9$ are for solid, solid circle, and solid square curves. The following parameters have been used: $\omega_{0}=1$, $\omega_{M}=\omega_{0}$, $\omega_{L}=30\;\omega_{0}$, $g=0.7\;\omega_{M}$ and $\gamma_{R}=\gamma_{L}=\gamma_{M}=\gamma =0.002\;\omega_{M}$.

**Figure 5 entropy-24-00032-f005:**
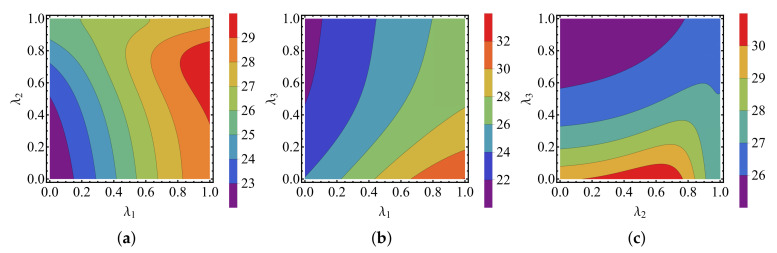
The amplification factor $\alpha_{L}$ dependence on the parameter $\lambda_{\alpha }$. In the figures, $\omega_{0}=1$, $\omega_{M}=\;\omega_{0}$, $\omega_{L}=30\;\omega_{0}$, $T_{L}=5\;\omega_{0}$, $T_{R}=0.5\;\omega_{0}$, $T_{M}=3\;\omega_{0}$, $g=0.3\;\omega_{M}$, and $\gamma_{L}=\gamma_{M}=\gamma_{R}=\gamma =0.002\;\omega_{M}$. In addition, (**a**) $\lambda_{3}=0.3$, (**b**) $\lambda_{2}=0.3$, and (**c**) $\lambda_{1}=0.3$.

**Figure 6 entropy-24-00032-f006:**
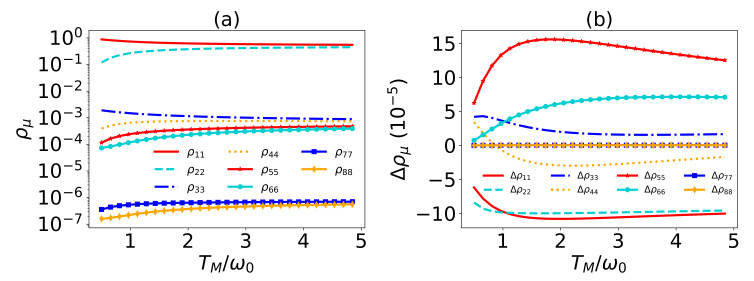
(**a**) The populations versus the temperature $T_{M}/\omega_{0}$. (**b**) The population difference between $\lambda_{1}=0.9$ and $\lambda_{1}=0$ versus the temperature $T_{M}$. In both figures, $g=0.7\;\omega_{M}$, $\lambda_{2}=\lambda_{3}=0$, $\omega_{0}=1$, $\omega_{M}=\;\omega_{0}$, $\omega_{L}=30\;\omega_{0}$, $T_{L}=5\;\omega_{0}$, $T_{R}=0.5\;\omega_{0}$, and $\gamma_{L}=\gamma_{M}=\gamma_{R}=\gamma =0.002\;\omega_{M}$.

**Figure 7 entropy-24-00032-f007:**
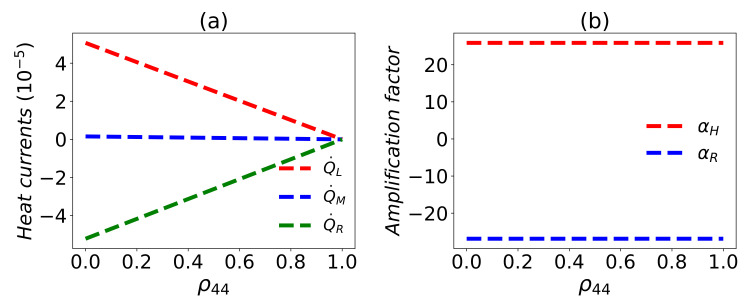
(**a**) Heat currents and (**b**) amplification factor versus $\rho_{44}=\rho_{44}\left(0\right)$. Here $\omega_{0}=1$, $\omega_{M}=\;\omega_{0}$, $\omega_{L}=30\;\omega_{0}$, $T_{L}=5\;\omega_{0}$, $T_{R}=0.5\;\omega_{0}$, $T_{M}=\omega_{0}$, $g=0.3\;\omega_{M}$, $\lambda_{1}=\lambda_{2}=\lambda_{3}=1$, and $\gamma_{L}=\gamma_{M}=\gamma_{R}=\gamma =0.002\;\omega_{M}$.

**Figure 8 entropy-24-00032-f008:**
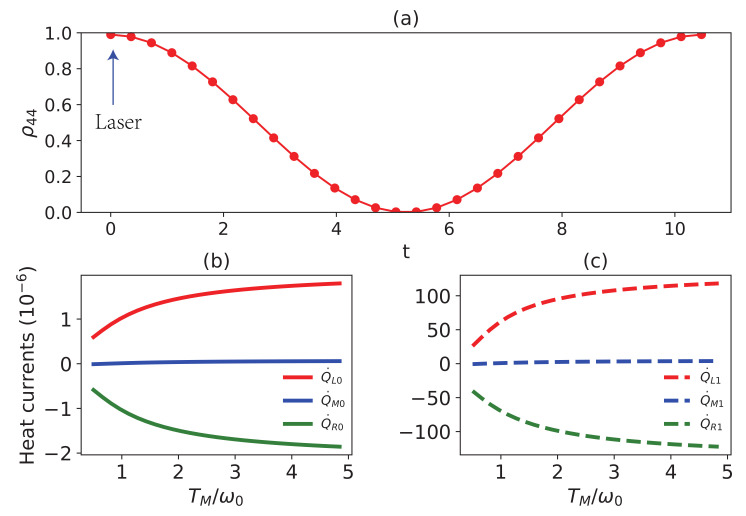
(**a**) $\rho_{44}$ versus the time *t*. The steady-state heat currents versus $T_{M}/\omega_{0}$ before (**b**) and after modulation (**c**). $\omega_{0}=1$, $\omega_{M}=\omega_{0}$, $\omega_{L}=30\;\omega_{0}$, $T_{L}=5\;\omega_{0}$, $T_{R}=0.5\;\omega_{0}$, $T_{M}=3\;\omega_{0}$, $g=0.7\;\omega_{M}$, and $\gamma_{L}=\gamma_{M}=\gamma_{R}=\gamma =0.004\;\omega_{M}$, $\Omega =0.3\;\omega_{0}$, and $\lambda_{1}=\lambda_{2}=\lambda_{3}=1$. $\rho_{44}=0.99$ in (**b**).

## Data Availability

Not applicable.

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
