# Peer review of "Common Environmental Effects on Quantum Thermal Transistor"

_entropy, 2021, doi:10.3390/e24010032_

Round 1

Reviewer 1 Report

The authors analyze the therml transistor effect in a system formed by three qubits (two-level systems or spins) with a three-body interaction and coupled to three external environments with linear (one-spin) interaction and with nonlinear (two-spin) interaction. They clearly show that a such system operates as a thermal transistor with a high amplification coefficient, as figure of merit. 

I found the work interesting and I recommend it for publication. But I would like that the authors addres my question which are reporte as follows.

1) In the title, they use the word "quantum"  and in the introduction they give an overview of papers related to the "Quantum Thermodynamics" (QT). However, they discuss a rate equation involving the populations of the states. A priori, a rate equation can also describe a classical system with discrete variables (0 and 1). The "quantum"  in this work (as well as in many other works of the scientific community working in QT) refers to the "energy quantization" that can characterize low-dimensional (nanoscale system) or other mesoscopic nanodevices. Can the authors make a comment about my observation? can they cite the references in which pure quantum coherence effect (i.e. off-diagonal element of the system density matrix) play a role in the thermodynamical quantities?

2)The authors discuss the results by varying the parameters related to the dissipative interaction with the environment. However, other parameters seem to be kept fixed in the overall discussion. For example, the qubits frequency ratio is set to 1:30:31 everywhere or the coupling constant g is 0.1, 0.3 or 0.7.  How do the result change in a different regime of these parameters? Can the author explain the reason of their choice, namely why they focus on these numbers? Can you make a comment?

3) Is it possible to give an upper bound of the maximum amplification factor which can be reached by varying all possible parameters (including frequency qubits and coupling interaction). Or at least, can you comment on this issue?

4) In the last part, it is shown that the state numberd "4" is a dark state and the steady state properties depend on the initial value for its occupation. This means that the system is non-Markovian by definition as it has memory of the initial state and the environment does not erase completely the initial state. In the framework of the theory of the open quantum systems, one should go beyong the Born-Markov approximation and must consider next-order effect which can lead to the fact that the nominal "dark state" is not really dark but it is coupled to the other states via next-order processes. Can the author make a comment about this issue?

Minor corrections

  • after Eq. 14, line 66, it should be written "in given in C" instead of "D", if I understood correctly
  • partial derivative in Eq.15 should be specificied respect to which quantity (temperature, I guess).
  • The figure in the appendix D shoud be labelled in a different way, not as "Fig.1" but as "Fig1A" in order to avoid the confusion of the figure 1 of the main text.

Reviewer 2 Report

Report on: Common environmental effects on quantum thermal transistor

Authors:
Yu-qiang Liu , Deng-hui Yu , Chang-shui Yu * 

The paper presents a well written overview about a fascinating development to use quantum effects in thermal transport for a transistor. I read the paper with great benefit and recommend it to publish. The model is described and the relevant informations provided. The authors discuss as new findings various possibilities to enhance the performance which shows that the manifold of dependencies give plenty space for new developments. As some suggestions may I add:
1.
Some critical remarks about the validity of Lindblad master equations could be in order. Strictly speaking, as shown in the literature it is only valid for harmonic oscillators.
2.
Before eq. 12 the modelling -though present in literature- could be a little bit enhanced to provide a concise readable picture. The remark of validity of secular equation and time scales is important to explain.
3.
Beginning of chapter 4, perhaps the definition of weak heat current in contrast to the right and left ones could be explained.
4.
Fig 4 caption: it is meant probably left/right figure instead upper/lower panel?
5.
In appendix C after eq (A6) the numbering starts again with (7) and the figure 1 is repeated and could be avoided. 

Reviewer 3 Report

The manuscript reported a theoretical study regarding the common environmental effect on quantum thermal transistor. The authors reported that the functions of the thermal transistor can controlled by the skillfully designed common environments. In particular, the dark state can provide an additional external channel to control the heat currents without any disturbance of the amplification rate in some conditions. The study is interesting and shows some new way to improve the performance of quantum thermal transistor. But I feel the manuscript has not been well organized therefore I would suggest some modifications before the publication. The followings are my concerns.

First, I think there should be some sentences to explain what it is common environmental effect. Even though readers can get it for references, it would be very helpful to explain it in the draft.

Second, when talking about ρii, what do they represent? Can we understand it as three two levels system plus one environment? I think this is also the key to understand the study. I would like the physics behind these equations to be explained more clearly.

Third, Fig 2,3,4,7, and 8 need to be modified. The unit of x-axis is not right. Delete the box line for the legend in the figure. On Fig. 3, the all the line should be labeled otherwise it is not understandable.

Reviewer 4 Report

The authors study a prototypical model of a quantum thermal machine composed by three qubits coupled to three thermal reservoirs, which can act as a thermal transistor (amplification of a heat current by increasing another one). The authors focus in the situation where the thermal reservoirs can be common, inducing collective dissipation, as controled by the parameters "lambda" in their respective interaction Hamiltonians, Eqs.(4).

Such model with "skillfully designed common environments" (to employ the authors' wording), has been originally introduced in Ref.[37], where it was already shown that the use of common environments can boost cooling by enhancing the heat currents in the model, including the case in which a dark state is obtained. This important fact (given that all this work is entirely based in the model of [37]) is not even acknowledged in the manuscript. Not stating clearly that the model here has been previously designed and studied (pointing to the correct reference) but simply changing its name to "thermal transistor" is a serious act of misconduct,  unacceptable in academia, and that may constitute plagiarism in my opinion. This point is accentuated because the authors try to present conclusions and findings of Ref.[37] as if they were a product of their research. For example in the conclusions (sec. V) it is said "It is found that the amplification rate of the transistor can be raised by properly designing the common couplings with the environments." where this "proper design" is from [37], and later on "When the system is coupled with the reservoirs with completely correlated transitions, a dark state will occur. This dark state can provide an additional channel to control the magnitude of the heat currents..." an important effect that was previously obtained in [37] as well.

The only two places where Ref.[37] is mentioned in the manuscript is (i) before Eqs.(4) (these equations are indeed original from [37]) but the reference is mixed with [81] to try to dilute it, and (ii) In page 5. where the authors state "We’d like to emphasize that the enhancement of the amplification rate occurs in the sense that all the parameters but the common couplings are kept invariant, which is different from the roles presented in Ref. [37]" a sentence which is not even true, since the case of keeping all the parameters invariant but the common couplings is also considered in [37].

Finally, and this is probably the reason why the authors tried to hide [37] and reapropriate from their results, I find that the original results presented in this manuscript are very weak. Since the model was already designed, the equations for the heat currents calculated and the dynamics explored in Ref.[37], and the idea of the thermal transistor already developed in Refs.[85, 49], the merit of the authors reduces to numerically evaluate Eq. (15) for different sets of parameters to find that, as somehow expected, the thermal transistor effect is also enhanced.

Round 2

Reviewer 4 Report

I think the article improved with the changes made by the authors. Reference [37] is now correctly acknowledged, and the novel part of this work (the strong coupling between the qubits) better emphasized, including the explanations on how the authors obtained the dynamics of the system in the present case.

Although the authors should have correcly acknowledged previous references crucial for their study from the beggining (I was wondering what would have happened if none of the referees were aware from [37]), I think the ammended manuscript is now acceptable, and could be of interest for the community working on quantum thermodynamics.

Therefore, I think the manuscript is now suitable for publication in Entropy.